# Heat Stress Resistance Mechanisms of Two Cucumber Varieties from Different Regions

**DOI:** 10.3390/ijms23031817

**Published:** 2022-02-05

**Authors:** Bingwei Yu, Fangyan Ming, Yonggui Liang, Yixi Wang, Yuwei Gan, Zhengkun Qiu, Shuangshuang Yan, Bihao Cao

**Affiliations:** 1Key Laboratory of Horticultural Crop Biology and Germplasm Innovation in South China, Ministry of Agriculture, College of Horticulture, South China Agricultural University, Guangzhou 510642, China; 20181017004@stu.scau.edu.cn (B.Y.); july@stu.scau.edu.cn (F.M.); a18077587209@stu.scau.edu.cn (Y.L.); wangyixi@stu.scau.edu.cn (Y.W.); gyw@stu.scau.edu.cn (Y.G.); qiuzhengkun@scau.edu.cn (Z.Q.); 2Guangdong Vegetable Engineering and Technology Research Center, South China Agricultural University, Guangzhou 510642, China

**Keywords:** heat stress, thermotolerance, transcriptome, hormone, transcription factors

## Abstract

High temperatures affect the yield and quality of vegetable crops. Unlike thermosensitive plants, thermotolerant plants have excellent systems for withstanding heat stress. This study evaluated various heat resistance indexes of the thermotolerant cucumber (TT) and thermosensitive cucumber (TS) plants at the seedling stage. The similarities and differences between the regulatory genes were assessed through transcriptome analysis to understand the mechanisms for heat stress resistance in cucumber. The TT plants exhibited enhanced leaf status, photosystem, root viability, and ROS scavenging under high temperature compared to the TS plants. Additionally, transcriptome analysis showed that the genes involved in photosynthesis, the chlorophyll metabolism, and defense responses were upregulated in TT plants but downregulated in TS plants. Zeatin riboside (ZR), brassinosteroid (BR), and jasmonic acid (JA) levels were higher in TT plants than in TS. The heat stress increased gibberellic acid (GA) and indoleacetic acid (IAA) levels in both plant lines; however, the level of GA was higher in TT. Correlation and interaction analyses revealed that heat cucumber heat resistance is regulated by a few transcription factor family genes and metabolic pathways. Our study revealed different phenotypic and physiological mechanisms of the heat response by the thermotolerant and thermosensitive cucumber plants. The plants were also shown to exhibit different expression profiles and metabolic pathways. The heat resistant pathways and genes of two cucumber varieties were also identified. These results enhance our understanding of the molecular mechanisms of cucumber response to high-temperature stress.

## 1. Introduction 

Extreme climate changes threaten agriculture and food safety. High temperature (HT) stress limits plant growth and productivity and even causes deaths under extreme conditions [1,2]. In the tropical and subtropical regions, vegetable crops, such as tomato, pepper, and cucumber, had reduced fruit number, weight, and morphology during spring and autumn due to global warming [3,4]. Moreover, the high temperature environment will also aggravate the impact of abiotic stresses, such as drought stress on plants, and induce the outbreak of a series of diseases [5,6]. Thus, a proper understanding of the molecular mechanisms of plant thermotolerance is needed for mitigating the negative effects of HT on crops. 

Cucumber is an annual vine originating from the foothills of the Himalayas [7]. Natural and artificial selection has contributed to the genetic differences observed between the cultivated and wild cucumber varieties [8]. Currently, China is the largest cucumber producer, contributing to more than two-thirds of the global output, followed by the European Union, Turkey, the United States, and other countries. 

Cucumber was introduced into China from Central Asia two thousand years ago. Two cucumber varieties are currently cultivated in China, the North China type (located in the northern parts of China’s Yangtze River) and the South China type (distributed in the southern areas of China). The North China type of cucumber is a club-shaped green fruit covered with dense white spines and grows well in wet and cool conditions. Conversely, the South China type of cucumber is cylindrical with a reticulated green and white coloration with sparse black spines and prefers humid and hot environments. 

Although there are some regional differences in cucumber varieties, in general, it is a crop that prefers a warm and cool environment, and the suitable temperature for growth is 20–30 °C. Above 35 °C will make it grow abnormally [9,10]. When the cucumber is damaged by heat, the leaves droop and turn yellow, the flowers fall easily, the fruit develops deformities. In more severe cases, the leaves are scorched and wilted by the heat, the flowers and fruits wither, the top dies, and the whole vine can die. 

Plants have evolved various response mechanisms to elevated temperatures. Physical changes and metabolism signals, including leaf orientation change, reduced water loss, membrane lipid composition alteration, and larger xylem vessels, are the common response mechanisms by plants to HT [11,12]. HT induces the production of reactive oxygen species (ROS), which causes the peroxidation of membrane and pigments, leading to membrane permeability loss [13]. In addition, HT alters the chloroplast and metabolite composition of leaves, thereby, reducing the photosynthetic rate and resulting in a short life cycle and diminished productivity in plants [14]. 

Extreme HT causes rapid cellular damage or death, which leads to catastrophic alterations of cellular organization [12,15]. Transcriptomic and proteomic analyses have revealed that HT induces various physiological responses and biotic and abiotic stress-related genes in plants. Such physiological responses include lipid and secondary metabolisms, calcium signaling, protein phosphorylation, phytohormone signaling, RNA metabolism, and transcription regulation [9,16,17,18,19,20]. 

Among these, transcription factors play an important role, particularly heat shock factors (HSFs), which are key regulators of HT responses in plants. *Heat shock transcription factor A1* (HSFA1) responds to HT by a series of transcriptional regulatory responses of thermotolerance in plants [12]. Furthermore, the *basic leucine zipper* (bZIP) gene family is reported to be involved in the HT response by endoplasmic reticulum-unfolded proteins [18,21]. The *basic helix-loop-helix* (bHLH) transcription factor, *phytochrome interacting factor 4 (PIF4),* has been demonstrated to control plants’ morphological acclimation during HT [22]. 

Under HT stress, the phytohormone signaling pathway-associated defense responses are stimulated [23,24]. Previous studies showed that the abscisic acid (ABA), salicylic acid (SA), and ethylene (ET) levels increased, while the cytokinin (CK), auxin, and gibberellic acids (GAs) levels decreased in response to HT. These fluctuations ultimately caused premature plant senescence [24,25,26]. Nonetheless, HT promotes auxin accumulation and stimulates GA and brassinosteroid (BR) pathways in hypocotyl elongation [27,28,29]. 

This study investigates the HT tolerance of two cucumber varieties from different regions. We measured the physiological indexes and endogenous hormones and systematically analyzed the genetic responses to heat stress. Distinct heat stress response patterns in transcription and translation regulation, hormone signaling pathways, and vascular patterns were observed between thermotolerant and thermosensitive cucumber plants. We also identified several TFs and hormones crucial for the HT response.

## 2. Materials and Methods

### 2.1. Plant Materials

The thermotolerant ‘TT’ and thermosensitive ‘TS’ cucumber inbred lines were grown in the standard culture chamber at South China Agricultural University (Guangzhou, China). The seeds were soaked in warm water at 55 °C for 20 min and 35 °C for 4 h, then they were wrapped in a damp cloth and germinated in an incubator at 28 °C. After sprouting neatly, they were planted in a sterilized fresh soil substrate and cultivated in a culture room at 25 °C/16 h-daytime and 20 °C/8 h-night. Water and pest control were provided according to standard protocols. All experiments were performed when the fourth true leaf of the seedlings began to unfold, and the same tissue site was used.

### 2.2. High-Temperature Stress Treatments, Chlorophyll Fluorescence and Root Viability

Tow temperature detectors were placed in greenhouses and the farmland next to it, with a height of 1.5 m, and temperature values were recorded every 15 min. Temperature monitoring was conducted from May to October to obtain an overview of the cultivation environment during the hottest period in Guangzhou. Therefore, the HT treatment was set as 43 °C for 16 h in the daytime and 25 °C for 8 h in the night for our experimental conditions. The recovery conditions were the same as for the normal culture environment. 

Chlorophyll was examined using the chlorophyll fluorescence imager (IMAGING-PAM, WALZ, Effeltrich, Germany) after 0, 3, and 6 days post HT treatment [9]. Dihydrorhodamine 123 (DHR123), a reactive oxygen species fluorescent probe was used for the detection of ROS [30]. Fluorescein Diacetate (FDA), a cell-permeable esterase substrate can only be broken down into fluorescein and accumulate in living cells. Propidium iodide (PI) is a cell-membrane impermeable dye. FDA/PI were used for the cell viability assessment [31]. 

The roots were soaked in phosphate-buffered saline (PBS) (pH 7.0) containing 1% (*w*/*v*) FDA and 0.5% (*w*/*v*) PI for 20 min in darkness and rinsed in PBS buffer for 10 min. For ROS detection, the roots were soaked in PBS (pH 7.0) buffer containing 50 mmol/L DHR123 for 20 min in the dark and rinsed in PBS buffer. The roots were then photographed using a fluorescence microscope (Carl Zeiss LSM710) with the specifications (488/530 nm), (535/615 nm), and (488/525 nm) for FDA, PI, and DHR123, respectively.

### 2.3. Determination of Physiological Indicators

The leaf tissues were subjected to a reaction with thiobarbituric acid (TBA) to measure the malonaldehyde (MDA) content [28]. Nitroblue tetrazolium and guaiacol reactions were used to determine the enzymatic activities of SOD and POD [29]. The relative electric conductivity of the leaf exudate was determined using a conductivity meter (DDS-307). Chlorophyll was extracted from fresh leaves (0.1 g) using 96% ethanol (5 mL) in the dark. Chlorophyll a and b (Chl a and b) and carotenoid were quantified spectrophotometrically at the wavelengths of 470, 649, and 665 nm, respectively. All physiological indexes were determined in nine biological replicates.

### 2.4. RNA Extraction, Library Construction, and RNA Sequencing (RNA-Seq)

Leaves from the TT and TS lines were collected after 0 (CK), 3, and 6 days post HT treatment for RNA-seq analysis. A sample constituted of leaves was pooled from ten plants. The total RNA was extracted using a C HiPure Plant RNA Kit (R4151, Megen, Guangzhou, China). Subsequently, messenger RNA (mRNA) was enriched using oligo (dT) beads, and the ribosomal RNA (rRNA) was removed using the Ribo-ZeroTM Magnetic Kit (Epicentre). 

The enriched mRNA was fragmented into shorter fragments using a fragmentation buffer and reverse-transcribed into complementary DNA (cDNA) with random primers. Thereafter, a buffer containing DNA polymerase I, RNase H, and dNTPs was used to synthesize the second cDNA strand. The cDNA fragments were purified, end-repaired, and polyadenylated. The fragments were then ligated to the Illumina sequencing adapters and sequenced using Illumina HiSeqTM 2500 by GeneDenovo Biotechnology Co. (Guangzhou, China).

### 2.5. RNA-Seq Data Analyses

Novel gene transcripts were identified and annotated using Cufflinks software [32] following read normalization and sequence alignments [33,34,35]. Gene identifications (IDs) corresponding to the gene symbols used in the article are in Appendix A. The gene abundances were quantified using RSEM software [36]. Additionally, correlation and principal component analysis were used to evaluate the relationship between samples. The edgeR package (11 March 2020, http://www.rproject.org/) was used to identify differentially expressed genes (DEGs) across samples. The DEGs represent genes with a fold change ≥2 and a false discovery rate (FDR) < 0.05. 

Enrichment analysis of the DEGs was conducted based on the Gene Ontology (GO) (1 April 2020, http://www.geneontology.org/) and Kyoto Encyclopedia of Genes and Genomes (KEGG) databases (2 April 2020, https://www.kegg.jp/kegg/) [37]. The expression pattern of DEGs was normalized to 0, log2 (v1/v0), and log2 (v2/v0) and then clustered to different profiles using the Short Time-series Expression Miner (STEM) software (Ernst et al., 2006). The clustered profiles with a *p*-value < 0.05 were considered significant. The clustered DEGs were then subjected to another round of enrichment analysis of the GO and KEGG pathways.

### 2.6. Quantitative Real-Time RT-PCR

All qRT-PCR analyses were performed with gene-specific primers (Appendix A) using a ChamQTM Universal SYBR qPCR master mix (Q711-02/03, Vazyme, Nanjing, China). Gene expression was assessed using the delta-delta Ct (2^−^^△△^^Ct^) method [38]. 

### 2.7. Determination of the Endogenous Hormones and Predictive Analysis of the Protein-Protein Interactions

Fresh plant samples (1 g) were extracted using cold methanol and later concentrated and purified. Endogenous hormones were determined by liquid chromatography-tandem mass spectrometry (LC-MS/MS) [39,40,41]. The KEGG database was used to enrich the DEGs of plant hormone signaling pathways. Furthermore, protein interactions between the transcription factors (TFs) and genes of the phytohormone signaling pathways were analyzed through the JASPAR software (16 August 2020, http://jaspar.genereg.net/). Interactions with the correlation coefficient > 0.9 and FDR < 0.05 were considered significant. Protein prediction was conducted based on the DEGs and TFs of the two cucumber varieties, and the KEGG database was used to enrich the related genes. The results were visualized using TBtools, Cytoscape, and Adobe illustrator.

## 3. Results

### 3.1. The Phenotype of Thermotolerant and Thermosensitive Cucumber Plants under High-Temperature Stress

Cucumbers prefer a humid and warm climate; however, some varieties can adapt to different climatic latitudes. A previous heat resistance study on the cucumber cotyledon stage showed that the inbred lines TT (South China thermotolerant cucumber) and TS (North China thermosensitive cucumber) had contrasting heat resistance (Figure 1A). In order to examine the heat resistance of the cucumber seedling stage, we assessed the summer temperatures of Guangzhou for several months and developed a one-week temperature curve (Figure 1B). 

The temperatures usually rise rapidly from about 23 °C at 6 a.m. (about 1 h after the sun rises above the horizon) to about 40 °C after 3 h during the summer in Guangzhou. High temperatures above 38 °C were sustained for 5.25–10.75 h during the day in summer, and the maximum greenhouse temperature was 2–5 °C lower than the farmland. The maximum farmland temperature was 54.5 °C, and the maximum temperature of greenhouse reached 49.5 °C. However, from 10 a.m. to 3 p.m., the average temperature reached 43 and 38 °C in the farmland and greenhouse, respectively (Figure 1B).

The study examined the heat resistance of TT and TS cucumber plants at the three leaves stage (Figure 1C,G). After three days of high temperature (43 °C) treatment, the TS leaves showed brown margins, wilted, and died, while the TT plants were in good condition with normal, green leaves. The leaves of TS drooped downward, while the TT leaves rose upward (Figure 1D–F). The angle between the petiole and blade of TT were larger than TS (Figure 1L,M), and this phenotype became severe after 6 days of high temperature (Figure 1L,M). Three days after recovery in a normal environment, nearly all TT plants were in good condition, while approximately 80% of the TS plants died (Figure 1K).

To further understand the physiological modifications in response to high temperature, some major physiological changes were measured. Malondialdehyde (MDA) is an important indicator of membrane lipid oxidation. The base MDA contents were different in TT and TS plants under normal conditions, but during heat treatment, TS contained high MDA content until the temperature returned to normal. The TT plants had reduced MDA contents, and after 6 days of heat treatment, their MDA content normalized (Figure 1N). 

Electrolyte leakage is an important indicator of cell membrane injury. The TS plants had high relative conductivity after high-temperature stress and failed to recover after the temperature returned to normal (Figure 1O). Superoxide dismutase (SOD) and peroxidase (POD) are two necessary antioxidant enzymes that protect plants from heat-induced oxidative stress. The base SOD and POD activities differed in TT and TS plants under normal conditions; however, TT plants restored enzyme activity levels during heat treatment (Figure 1P,Q). These results indicate that the thermotolerant cucumber inbred line TT has enhanced physical and physiological adaptation to high-temperature stress.

### 3.2. Photosystem of Thermotolerant and Thermosensitive Cucumber in Response to High-Temperature Stress

The study observed the leaf chlorophyll fluorescence and photosynthetic system parameters to explore the photosynthetic system of the two cucumber lines under high-temperature stress. The TS plants had a more active photosystem than TT plants in the normal environment (Figure 2A,D). However, TT plants showed stable photosynthesis in all the measured leaves after 3 days of high-temperature stress and rapid recovery in the normal growth environment. TS plants had extremely unstable photosynthesis after 3 days of high-temperature stress (Figure 2A–F). 

The Fv/Fm curve (maximum quantum yield of PS II photochemistry) showed that TS and TT plants have similar photosynthetic capacities under normal light, smothered under strong light (Figure 3G). High temperature seriously impaired the photosynthetic system of TS plants but slightly impaired the TT photosynthetic system (Figure 3H). 

Under a normal environment, the ETR (relative electron transfer rate) curve of TS was higher than TT plants. Still, the photosynthetic capacity of TT plants increased, while that of TS plants decreased after 3 days of high-temperature treatment (Figure 2I,J). The NPQ (non-photochemical quenching), Y(NPQ), and qN curves, which represent the ability of a plant to dissipate excess light energy, demonstrated a photo protective ability that was stronger in TT plants than in TS plants. 

Moreover, high temperatures weakened the photosynthesis capabilities of TT and TS plants (Figure 2M,N, Appendix A). The qL and qP curves indicated that TT plants maintained a stronger light quenching ability than TS plants after high-temperature stress. Thus, TT plants may maintain a normal activity of the PSII center in high temperatures (Appendix A). The Y (II) curve (effective quantum yield of photochemical energy conversion in PS II) was higher in TT than in TS (Figure 2M,N). 

This curve estimates the effective portion of absorbed quanta used in PSII reaction centers. Under normal environments, TT and TS plants have identical total chlorophyll, chlorophyll a, chlorophyll b, and carotenoids (Figure 2O–R). However, high temperature reduced all the chlorophyll parameters, and the damage was irreparable in TS plants after returning the plants to the normal environment. In contrast, the high temperature decreased the total chlorophyll, chlorophyll a, and carotenoids in TT, but the plants recovered rapidly after returning to the normal environment (Figure 2O–R). 

### 3.3. Cell Viability and Reactive Oxygen Accumulation in Thermotolerant and Thermosensitive Cucumber under High-Temperature Stress

We further observed the cell viability and oxygen accumulation in the root and stem. The TT plants showed enhanced root activity under high temperatures compared with TS plants (Figure 3A–E’). Most cells in the TS root died after 3 days of recovery treatment, yet the TT roots recovered well (Figure 3A–F). Similarly, cells in the TS stem died, and returning to normal temperature for 3 days aggravated the cell death (Figure 3G–I). The TT stem retained numerous live vascular bundles (Figure 3J–L). 

In addition, TS roots accumulated more reactive oxygen compared with TT roots (Figure 3M–R). In the stem, reactive oxygen was concentrated in the xylem of vascular bundles of TT plants but spread all over the vascular bundles of TS plants (Figure 3S–X). In the soil layer, high temperature reduced the root quantity of TT, which recovered rapidly after returning to normal temperature; however, TS had difficulty returning to normalcy (Figure 3Y,Z). These results indicate that TT plants have stronger heat resistance and resilience. 

### 3.4. Thermotolerant and Thermosensitive Cucumbers Have Distinct Expression Profiles and Transcriptomic Differentiation under High-Temperature Stress

Transcriptome sequencing was conducted at 0, 3, and 6 days after treating TT and TS plants with high temperatures. High-temperature treatment significantly increased twice as many upregulated DEGs in TT compared with in TS plants (Appendix A). Hence, the two cucumber lines responded differently to heat stress. Quantitative real-time RT-PCR (qRT-PCR) analysis of 12 genes verified the RNA-seq identified DEGs (Appendix A). 

The cluster analysis revealed four significant expression patterns in TS after heat treatment, while TT plants had only three significant patterns. Genes in profiles 0 and 1 were down-regulated and upregulated at profiles 6 and 7 (Figure 4A). The significantly enriched (*p* < 0.05) pathways were extracted to understand heat resistance between the two varieties. A Venn diagram showed the correlation (Figure 4B). There were 58 significant pathways in TS and 52 in TT with 29 identical and 15 similar expression patterns. 

The same upregulated pathways in the two varieties under heat stress included peroxisome, protein processing in the endoplasmic reticulum, valine, leucine isoleucine degradation, and autophagy regulation. Moreover, the down-regulated pathways included ribosomes, the biosynthesis of amino acids, porphyrin, and the chlorophyll metabolism (Figure 4B). DNA replication, plant–pathogen interactions, and the glutathione metabolism showed different expression patterns (red stars in Figure 4B). 

High temperature enriched 23 and 29 pathways in TT and TS plants, implying varied responses in both varieties (Figure 4C,D). Ubiquitin-mediated proteolysis, terpenoid backbone biosynthesis, and RNA polymerase were specifically upregulated in TT plants (Figure 4C). However, the fructose and mannose metabolism, galactose metabolism, and cysteine and methionine metabolism were specifically upregulated in TS plants (Figure 4D). Seven and nine crucial physiological pathways were down-regulated in TT and TS plants, respectively. 

Venn clustering showed that high temperature upregulated 6826 and down-regulated 4111 genes in TT and TS plants (Figure 5A–C). Of these, 1140 upregulated and 415 down-regulated genes were common to both varieties (Figure 5A,B). Gene Ontology (GO) enrichment analysis revealed that the commonly upregulated genes control responses to stimulus, stress, and heat (Figure 5D). However, the common down-regulated genes were related to metabolic processes, cellular component organization, and development (Figure 5H). 

The 39 genes enriching the cell cycle, metabolic process, and phenylpropanoid biosynthetic process were upregulated in TT and down-regulated in TS (Figure 5G). However, high temperature specifically upregulated 2867 and 923 genes in TT and TS, respectively (Figure 5A). Moreover, the specifically upregulated genes enriched DNA repair, DNA metabolic processing, the protein modification process, the regulation of gene expression, the response to stress, and the xylem–phloem formation processes in TT plants (Figure 5E). In comparison, the upregulated genes significantly enriched the lipid metabolic processes, water homeostasis, and transport in TS plants (Figure 5F). 

High temperature specifically repressed 523 and 1888 genes in TT and TS, respectively (Figure 5B). Interestingly, down-regulated genes specifically enriched metabolic processes in TT (Figure 5I). The TS-specific down-regulated genes increased pathways in response to stress, cell development, and the organization process (Figure 5J). Altogether, the transcriptome profiles demonstrated that DNA and protein processes, the hormone signaling pathway, and gene expression regulation involve heat responses.

### 3.5. Hormones That Signal Transduction Pathways Were Different between Thermotolerant and Thermosensitive Cucumber

High temperature affected several hormone synthesis precursors and signal transduction pathways (Figure 4). Under the normal environment, TT and TS plants had similar levels of gibberellin (GA), jasmonic acid (JA), brassinolide (BR), and abscisic acid (ABA), while TT had less auxin (IAA) and more cytokinin (ZR) than did TS (Figure 6A–F). After high-temperature treatment, the contents of ZR, BR, and JA were higher in TT than in TS plants (Figure 6A–F). However, GA increased in both varieties but was higher in TT after high-temperature stress (Figure 6B). Heat treatment boosted IAA and GA in TT and TS (Figure 6C–F). 

The genes involved in the plant hormone signal transduction pathway were analyzed further to investigate the hormone functions in response to high temperatures. Heat stress more quickly upregulated *TIR1* in TT compared with in TS. Moreover, heat stress down-regulated most Aux/IAA proteins that repress auxin-responses in both varieties but activated auxin response factors (ARFs) (Figure 6G). High-temperature treatment also induced multistep cytokinin signaling and upregulated type-A-ARRs inhibitors of cytokinin signaling in TS (Figure 6G). 

Moreover, the downstream *CsCYCD3* family genes responsible for cell division were upregulated in TT and down-regulated in TS. GO analysis showed that genes involved in the ethylene metabolic process were upregulated in TT and TS plants (Figure 6G). Heat stress upregulated the GA receptor *CsGID1* and down-regulated the *DELLA* inhibitor (Figure 6G). Heat stress also upregulated the negative regulator, *CTR1*, in the ethylene pathway, but EBF1/2 had a higher expression in TS compared with in TT plants and targets *EIN3* for proteolysis (Figure 6G). 

High temperature-induced the binding factor *JAR1* of JA, while the repressor JAZ and transcriptional activation factor MYC2 responded quicker in TT than in TS (Figure 6G). Genes of BR biosynthesis increased in TT and decreased in TS after HT (Figure 4C,D). In addition, *CsBZR1*, a positive regulator of the BR signaling pathway, accumulated earlier and higher in TT than in TS (Figure 6G). Variety TS maintained high levels of the major activator, *CsNPR1*, in the SA pathway in normal high-temperature environments. In contrast, the *CsNPR1* expression was firstly raised and then reduced in TT after high temperatures. 

Likewise, heat stress activated the *NPR1*-bound *TGA*, which masks its repressor domain and activates transcription (Figure 6G). These results imply that phytohormones are important for the response to high temperatures in cucumbers.

### 3.6. Transcription Factors Regulate the Responsive Differentiation of Thermotolerant and Thermosensitive Cucumber to High Temperature

Transcription factors (TFs) and their direct targets are the core regulators of the high-temperature response. Under high temperature, the TFs expression patterns identified 1668 transcription factors in 57 different families, and heat stress significantly affected some of these (Appendix A). Venn clustering analysis was conducted (Appendix A). High temperature upregulated 281 TFs in at least one-time point of each plant (TT or TS). The common TFs included HSF, NAC, MYB, and ERF (Appendix A). 

Interestingly, heat significantly upregulated most TT genes. However, the same genes were insensitive to the heat response in TS, where no significant changes in expression occurred (Appendix A). There were 392 down-regulated TFs in at least one stage of each plant, and they showed different expression patterns in TT and TS plants (Appendix A). Several TFs, including WRKY, MYB, ERF, and NAC transcription factors, were specifically upregulated in TT and down-regulated in TS after HT (Appendix A). 

Protein interaction predictions between the plant hormone, signal transduction pathway, and TFs were conducted to find the cucumber strategies of coping with high temperature (Appendix A). Fifteen TF families demonstrated a significant relationship with plant hormone signal transduction. Among them, the transcriptional activation pathway of auxin highly correlated with transcription factor families, including WRKY, MYB-regulated, C2H2, NAC, and HSF. Multiple transcription factor families, such as bZIP, NAC, ZF-HD, MYB, and bHLH (Appendix A), regulated the auxin feedback regulation mechanism. 

The bHLH and NAC factors significantly affected abscisic acid regulation. The ERF family genes linked to ethylene, abscisic acid, auxin, and bZIP were also significant. MYB also affects cytokinin signal transduction and downstream BR activation. Moreover, the C2H2 family interacted with the repressor JAZ of JA and regulated the GA and ABA perception factors. The HD-ZIP family regulates SA signal perception and transcriptional activation (Appendix A). Altogether, vital transcription factors have complex interactions with plant hormones in response to high temperatures.

Therefore, this study constructed an interaction model that assembled metabolic pathways and TFs (Figure 7). Seven of the 27 metabolic pathways were common in thermotolerant and thermosensitive cucumbers, but 13 enriched pathways were specific in thermosensitive cucumbers, and six were only in thermotolerant cucumbers. The metabolic pathways for the energy metabolism, metabolism of amino acids, lipid metabolism, signal transduction, repair and catabolism, and biosynthesis of secondary metabolites. 

Heat stress specifically enriched the lipid metabolism, metabolism of amino acids, and biosynthesis of secondary metabolites in TS. Repair, catabolism, and the energy metabolism were specifically enriched in TT. Heat stress upregulated most transcription factor families, such as HSF, bZIP, NAC, ARF, and DOF, while the GATA family was down-regulated. The bHLH, WRKY, and ERF families had polarized expression (Figure 7, Appendix A). 

The HSF family, including *HSFA2-6*, regulating HSP expression and other factors, was upregulated after 3 and 6 days of high-temperature treatment in TT and TS (Appendix A). High temperature upregulated the bZIP, bHLH, HD-ZIP, NAC, and WRKY family genes in both TT and TS plants; however, TT had faster and higher expression (Appendix A). 

High-temperature stress gradually increased the expression of *CsBIM1* and *CsbHLH20* in the bHLH family, and *CsDREB2* and *CsSHN3* in the ERF family in TT, but the expression declined gradually in TS. CsPIF3 and CsWRKY7 expression increased in TT and then TS. Interestingly, the high temperature rapidly upregulated the KNOX family genes in TT, which function in leaf polarity corresponding to the leaf orientation. In general, a limited family of transcription factors affects specific metabolic pathways, differentiating cucumber heat tolerance. 

## 4. Discussion

Global warming is a critical and urgent global concern that is worth investigating as it influences how plants cope with elevated temperatures [42,43]. Plants use complex regulation mechanisms, which involve multiple interaction pathways to cope with stress. Some of these mechanisms include the regulation of plant hormones, transcription factors and miRNA; the transmission and influence of signal factors; as well as the production and accumulation of metabolites [44,45]. Many studies have described the physical response of plants to high-temperature stress; however, the response of cucumber to high-temperature stress has been rarely studied. 

In the present study, thermo-tolerant and thermo-sensitive cucumber species from different regions were selected to identify their physical responses to high-temperature exposure. Gene expression and physiological changes analyses were performed to examine the responses of the two cucumber varieties to heat stress following 3 and 6 days of exposure. The result showed that the two cucumber cultivars with huge geographical differences exhibited large morphological differences in the leaves, roots, and stems when exposed to high temperatures. 

Additionally, various physiological indicators of photosynthetic systems, such as the active oxygen, antioxidant enzymes, and membrane systems, showed significant differences. Thermo-tolerant cucumber species restore functions to a relatively healthy level under high-temperature exposure. A combination of the changes in endogenous hormone content and gene expression revealed that cucumber species with different heat resistance traits demonstrated different gene expression strategies in response to heat stress, which involved a small number of transcription factor families. 

### 4.1. Changes in Cucumber Structure and Organization under Heat Stress

When plants are exposed to high temperatures, sensitive plants show cellular metabolic imbalances, resulting in a damaged photosynthetic system and the accumulation of harmful substances in the roots, stems, leaves, flowers, etc. This accumulation subsequently impedes plant growth and development. However, restoring the normal growth environment did not effectively induce damage repair but further compromised the plant growth or induced plant death. Plants that survived the stress displayed a stronger vitality, insusceptible cellular homeostasis, normal photosynthesis, and resilience [18,46,47,48]. 

Physical phenotypes characterized by leaf etiolation and hyponasty in thermo-tolerant and thermo-sensitive cucumber plants show wide differences under high temperatures. Thus investigating the regulators that stabilize cellular metabolic processes in the thermo-tolerant plant is an efficient strategy for exploring the mechanisms underlying crop heat tolerance. Similar reports showed that the leaf angle and petiole length increased with temperature increase in *Arabidopsis thaliana* [49,50]. The photosynthetic systems of the two species of cucumber were damaged differently, which is common in crops. 

These differences in response to high-temperature environments by different photosynthetic systems may be related to the function of Rubisco, which is the key enzyme responsible for autotrophic carbon fixation and oxygen metabolism [14,51,52,53]. TT plants maintained normal growth under high temperatures due to their robust and stable photosynthetic system. There was rapid recovery in the total chlorophyll, chlorophyll a, and carotenoids, which is characteristic of green leaves in TT plants when returning to a normal environment. 

This finding corroborates reports in other crops [54,55]. Conversely, TS plants were thermo-sensitive due to the vulnerability of their photosynthetic system. Decreased cell viability and accumulation of reactive oxygen in the roots and stem may have resulted in the death of TS plants. 

Transcriptome analysis of TT and TS plants showed that there were obvious differences in peroxisomes [56], photosynthesis [57], plant–pathogen interactions [58], ubiquitin-mediated proteolysis [59], plant hormones [60], and pyruvate and other metabolic pathways [61], which function in the heat stress response. These results are consistent with the characterization of thermo-tolerant and thermo-sensitive cucumber in the present study. Notably, the synthesis and metabolism of energy materials and genes for phytohormone pathways had opposite responses to high temperature in TT and TS plants, which could be an alternative strategy by which thermo-tolerant plants survive in high-temperature environments.

### 4.2. Phenotypic and Endogenous Hormone Pathways in Response to High Temperature

Under high-temperature stress, genes responsible for protein modification, DNA repair, macromolecule metabolism, and other processes in vivo are specifically upregulated in thermo-tolerant cucumbers. This is the response of plants to self-protection and adaptation under external stress [62,63,64,65]. Plants adapt to heat injury through changing their architecture, characterized by the development of elongated hypocotyl, hyponastic leaves, and enlarged xylem vessels [12,42,49]. Notably, PIF4 is the central regulator for morphological acclimation [63,64]. 

However, auxin biosynthetic genes (*YUCs*), auxin-responsive genes (*IAAs*), and auxin target genes (*SAURs*) are the downstream targets of *PIF4* in warmth-induced hypocotyl and petiole elongation and leaf hyponasty [28,66]. Sun et al. (2012) reported that high temperature-mediated expression of *YUC8* occurred later than PIF4 in *Arabidopsis*. In cucumber, the expression of *CsPIF3* increased in thermo-tolerant plants when compared with thermosensitive plants. 

There were significant differences in the expressions of *CsYUCs*, *CsIAAs*, and *CsSAUR**s* between TT and TS plants. It was possible that *PIF3* was more relevant instead of *PIF4* in cucumber. The implication of these results is that auxin-mediated phenotypic changes were likely important for the high-temperature response of cucumber. *Brevipedicellus* (*BP*), a member of the KNOX family, is a key regulator of plant architecture in *Arabidopsis*. 

Similarly, it has also been reported that KNOX and YABBY family genes regulate vascular development in other plants [67,68]. In the present study, the expressions of several YABBY and KNOX family genes were differentially driven by high temperature, which may be the process involved in cucumber adaptation to high-temperature [69]. The variations observed in the leaf and vascular system between thermo-tolerant and thermosensitive plants suggest that YABBY- or KNOX-mediated plant architecture was involved in the heat stress response. 

Plant hormones play a complex role in plant stress responses. In this study, hormones and signal transduction pathways were significantly triggered following high-temperature exposure. In a similar study, [70] demonstrated that *Gast1 protein homolog 5* (*GASA5*), encoded the GA-stimulated regulator of seedling thermo-tolerance. Under heat stress, ethylene accumulated in tomato plants and improved pollen quality, but the mechanism for these effects remains is unclear [71,72,73]. Other studies also reported that ABA is a typical stress hormone [74,75]. 

In Arabidopsis, *Ascorbate peroxidase 6* (*APX6*) reduces reactive oxygen accumulation through ABA and auxin signal pathways, ensuring seed germination under heat stress [76]. JA functions as an inhibitory factor in the stress response by interacting with various hormones, such as auxin. Further, JA promotes programmed cell death by inducing ROS production and regulating leaf morphogenesis. It protects the plant’s fertility under heat stress, while the content, synthesis, and signal transduction of JA are inhibited under heat stress [77,78,79,80,81]. Auxins reduce stress injury caused by high-temperature exposure through improving pollen fertility, thus, improving the crop yield [82,83,84,85]. BR regulates the plant xylem differentiation and architecture in response to heat stress (Bajguz & Hayat, 2009; Vriet et al., 2012). 

It also plays an important role in promoting quick recovery after exposure to heat stress and reduced oxidative stress (Dhaubhadel et al., 2002; Filek et al., 2019; Nie et al., 2013). *Brassinzole resistant 1* (*BZR1*), a critical regulator of BR, regulates the heat stress response through *RBOH1*-dependent reactive oxygen species (ROS) signaling in tomatoes (Yin et al., 2018). In the present study, BR biosynthesis genes were observed with quicker and higher *BZR1* accumulation in TT plants. This finding suggests that the BR pathway may have played critical roles in maintaining vitality under high temperatures, which may explain the thermo-tolerant nature of TT plants. 

### 4.3. Transcriptional Regulation Strategies after Heat Stress in Cucumber

Transcription factors are important in regulating plant development and stress responses [86,87]. Although constitutively expressed TFs may play crucial roles in regulating gene expression under high temperatures, heat stress-induced or heat-suppressed TFs in thermotolerant and thermosensitive plants potentially contribute to the differential regulation of downstream genes. HSF is one of the earliest reported and most studied transcription factors family related to heat stress regulation in plants, such as Arabidopsis [88,89], rice [90], tomatoes [91], and potatoes [50]. 

However, the relationship between HSF and plant hormones in modulating heat stress has been rarely reported [92]. WAKY family genes function in the plant heat response, including *WRKY25*, *WRKY26*, *WRKY33*, and *WRKY39,* in Arabidopsis [93,94]; *WRKY106* in maize [95]; *WRKY1* and *WRKY33* in wheat [96]; and *WRKY6* and *WRKY40* in pepper [97]. In the present study, half of the WRKY family genes were shown to have opposite expression patterns in TT and TS plants. 

Transcription factors and plant hormones corporately regulate the stress response. For instance, the Arabidopsis WRKY genes promote root development under salt stress by regulating ABA signaling and auxin homeostasis and altering the plant’s sensitivity to salt stress by inhibiting auxin synthesis genes [98,99]. On the other hand, *bZIP73* and *bZIP71* regulate cold resistance by ABA and ROS homeostasis in rice [100], while *MYB7* inhibits seed germination by negatively regulating the bZIP transcription factor and interacting with the ABA signal pathway in Arabidopsis under salt stress [101]. 

In our study, high temperature upregulated the expressions of HSFs and transcription factors in ERF, NAC, WRKY, bZIP, and bHLH family genes. Moreover, these gene families regulate the response of many significantly different metabolic pathways to heat stress in two varieties. To buttress this finding, an interaction model that assembled metabolic pathways and TFs was drawn. All cucumber pathways significantly enriched by heat stress were divided into six: the energy metabolism, the amino acid metabolism, the lipid metabolism, signal transduction, repair and catabolism, and the biosynthesis of secondary metabolites. 

The lipid metabolism, amino acid metabolism, and biosynthesis of secondary metabolites were significantly enriched in thermo-sensitive cucumbers, while repair and catabolism and the energy metabolism were enriched in thermo-tolerant cucumbers. Heat stress upregulated most of the transcription factor families, such as HSF, bZIP, NAC, ARF, and DOF, but down-regulated a few families, such as GATA, bHLH, WRKY, and ERF. 

Notably, the expression patterns of these genes differed between the two cucumber varieties with different heat resistance capabilities. Therefore, the differential expression of these upstream transcription factors could have led to the different strategies for downstream metabolic pathways in coping with high-temperature exposure, thus, resulting in the different heat resistances of the cucumber varieties.

## Figures and Tables

**Figure 1 ijms-23-01817-f001:**
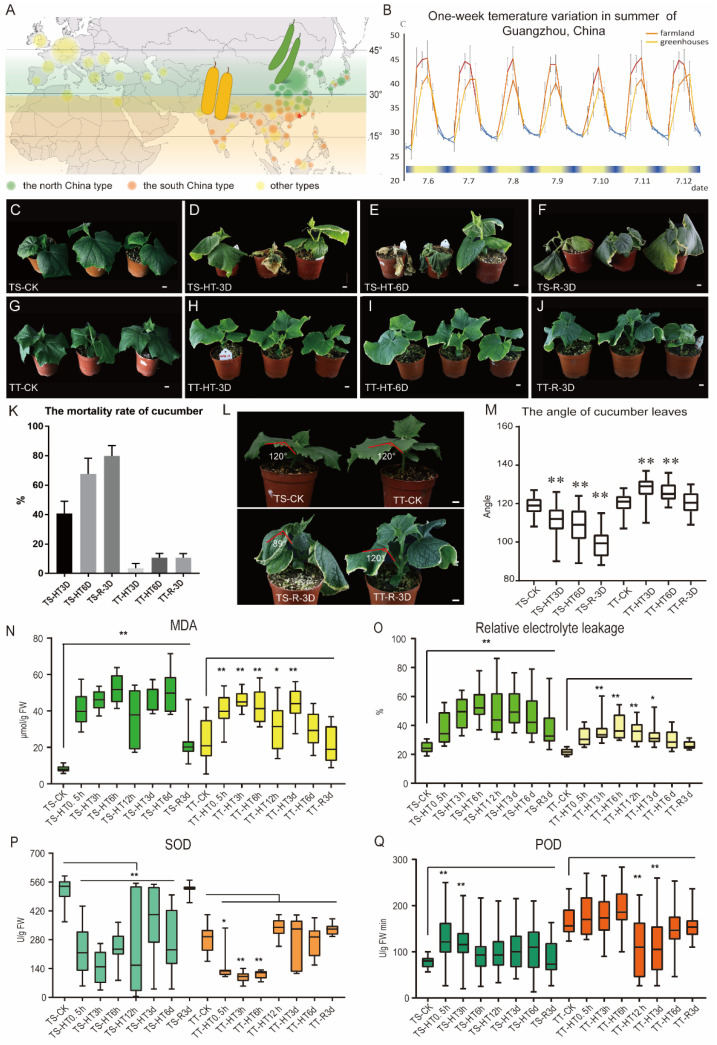
Phenotypic characterization of thermo-tolerant and thermo-sensitive cucumber under high-temperature stress. (**A**) Planting and distribution areas of the North China type TS and South China type TT. The asterisk position represents the experimental cultivation area (green and yellow), while the numbers represent the latitude. The size of the circle represents the approximate distribution of different types of cucumber varieties. (**B**) Temperature statistics of farming areas in Guangzhou, China, for a week in July. On the horizontal axis, blue represents night, while beige denotes day temperatures. The value of each point represents the average temperature for three hours. (**C**–**J**) Characterization of thermo-sensitive TS and thermo-tolerant TT lines. 3 days (**D**,**H**), 6 days (**E**,**I**), and recovered 3 days (**F**,**J**) (28 °C daytime/25 °C night) after high-temperature stress (43 °C daytime/25 °C night). TS and TT represent the thermo-sensitive and thermo-tolerant cucumber inbred lines. HT, high temperature; 3D, 3 day; 6D, 6 day; and R, recovered. (**K**) Death rate of 02 and 14 lines after high-temperature exposure. (**L**,**M**) Hyponasty angle between lamina and petiole after high-temperature exposure. (**N**–**Q**) Changes in the MDA (malondialdehyde) content, relative conductivity, SOD (superoxide dismutase), and POD (peroxidase) enzyme activities in thermosensitive TS and thermo-tolerant TT cucumber leaves after high-temperature stress. Scale bars represent 1 cm in (**C**–**J**,**L**). Values are the means ± SE (n = 30) in (**M**–**Q**). Relative to CK, ** in M-Q indicate significant differences of *p* < 0.01 by *t*-test, and * indicates *p* < 0.05 by *t*-test. FW, sample fresh weight; U/g FW, enzyme viability unit per gram of fresh weight; and U/g FW min, the enzyme activity unit per minute of fresh weight change. The % of relative electrolyte leakage indicates the relative electrolyte extravasation rate.

**Figure 2 ijms-23-01817-f002:**
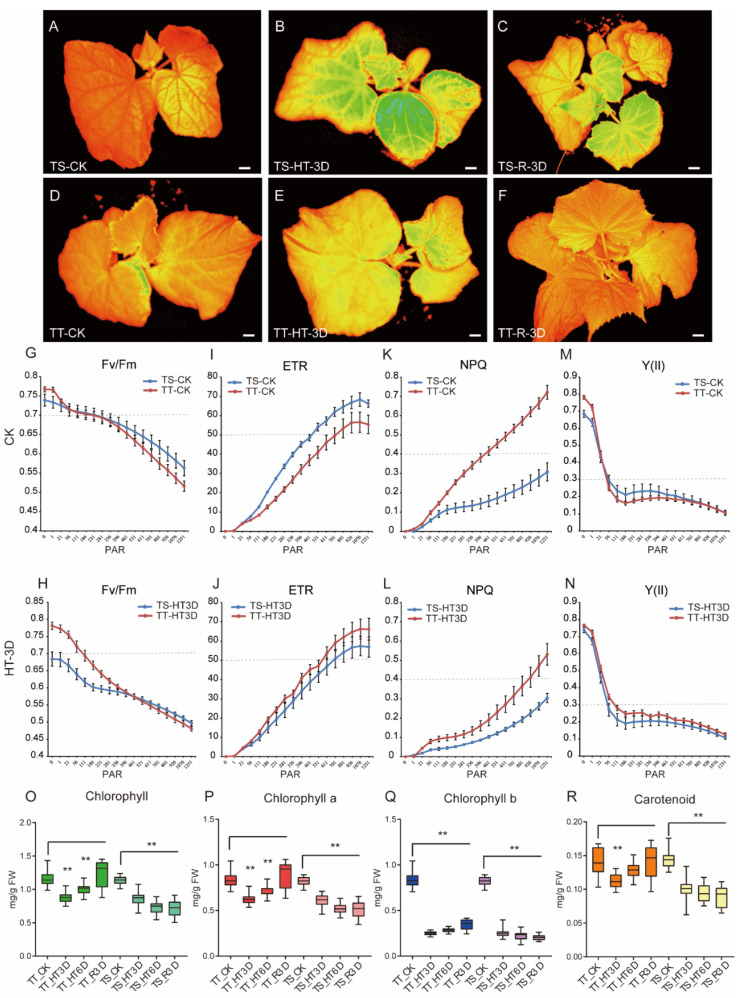
Chlorophyll fluorescence of thermo-sensitive and thermo-tolerant lines after high-temperature stress. (**A**–**F**) Chlorophyll fluorescence (Fv/Fm) of the TS and TT lines after high-temperature stress. The orange-red color indicates a stable photosynthetic system in leaves, while the green color indicates an unstable photosynthetic system. (**G**–**N**) The maximum quantum yield of PS II photochemistry (Fv/Fm) (**G**,**H**), the relative electron transfer rate (ETR) (**I**,**J**), non-photochemical quenching (NPQ) (**K**,**L**), the effective quantum yield of photochemical energy conversion in PS II (Y(II)) (**M**,**N**) of TS and TT lines after high-temperature stress. PAR denotes Photosynthetically Active Radiation. (**O**–**R**) The content changes of chlorophyll II, chlorophyll II a, chlorophyll II b, and carotenoids of TS and TT lines after high-temperature stress. Scale bars represent 1 cm in (**A**–**F**). Values are the means ± SE (n = 30) in (**G**–**R**). ** in O-R indicate significant differences of *p* < 0.01 by *t*-test, and one asterisk indicates *p* < 0.05 by *t*-test.

**Figure 3 ijms-23-01817-f003:**
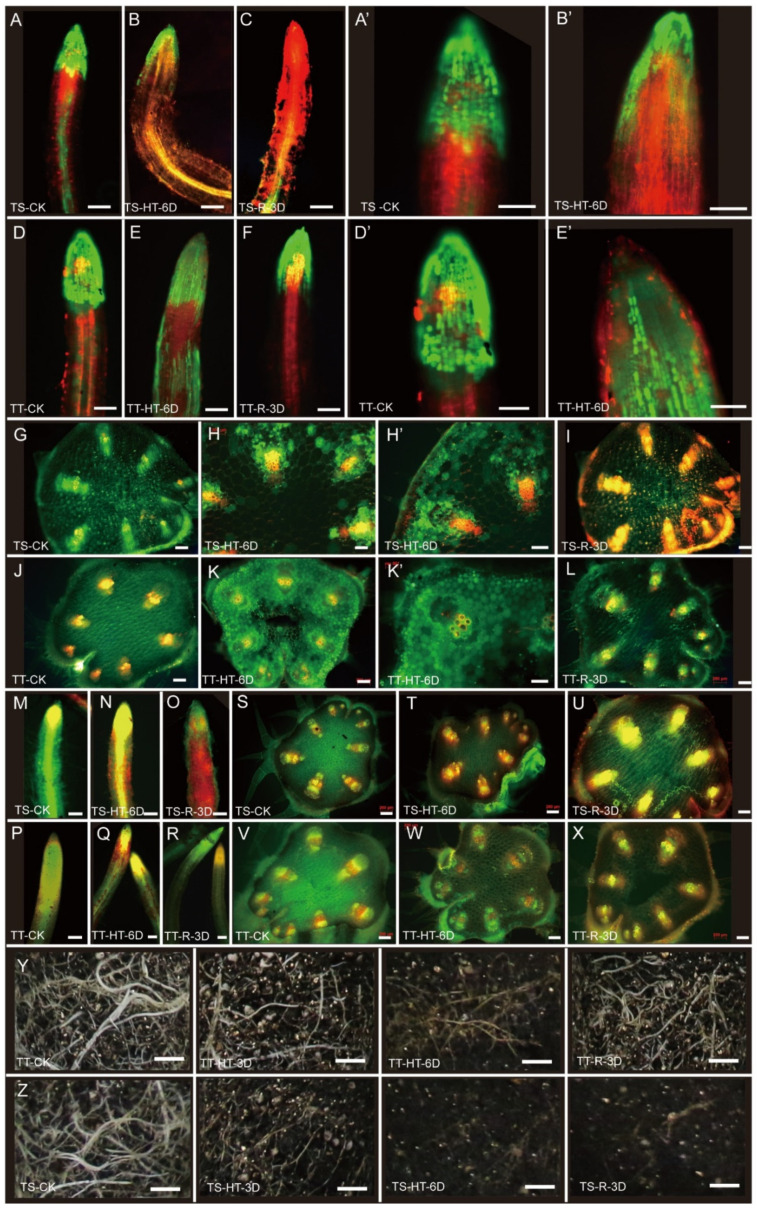
Cell viability and reactive oxygen accumulation in cucumber root and stem after high-temperature stress. (**A**–**F**), Characterization of cell viability in the root of TS (**A**–**C**) and TT (**D**–**F**) after high-temperature stress. (**A**’,**B**’,**D**’,**E**’) are amplifications of figures (**A**,**B**,**D**,**E**), respectively. Red fluorescence represents dead cells as revealed by PI (propidium iodide) staining, while the green fluorescence represents the living cells as revealed by FDA (Di-O-acetylfluorescein) staining. (**G**–**L**), the cell viability of the vascular system in the stem of TS (**G**–**I**) and TT (**J**–**L**) after high-temperature stress. (**H**’,**K**’) are amplifications of figures (**H**,**K**), respectively. Red fluorescence represents the dead cells stained by PI, while the green fluorescence represents the living cells dyed by FDA. (**M**–**X**), the characterization of reactive oxygen in the root (**M**–**R**) and stem (**S**–**X**) after high-temperature stress. The red fluorescence represents reactive oxygen concentrated region, the green is spontaneous fluorescence of plants, while yellow represents an overlap of red and green fluorescence. (**Y**,**Z**), the root system of cucumber in the soil layer for TS and TT plants after heat stress. TS and TT represent the thermo-sensitive and thermo-tolerant cucumber inbred lines. HT, high temperature; 3D, 3 day; 6D, 6 day; and R, recovered. Scale bars represent 100 µm in (**A**–**C**,**F**,**G**,**I**), 200 µm in (**D**,**E**,**H**,**J**), and 1 cm in (**Y**,**Z**).

**Figure 4 ijms-23-01817-f004:**
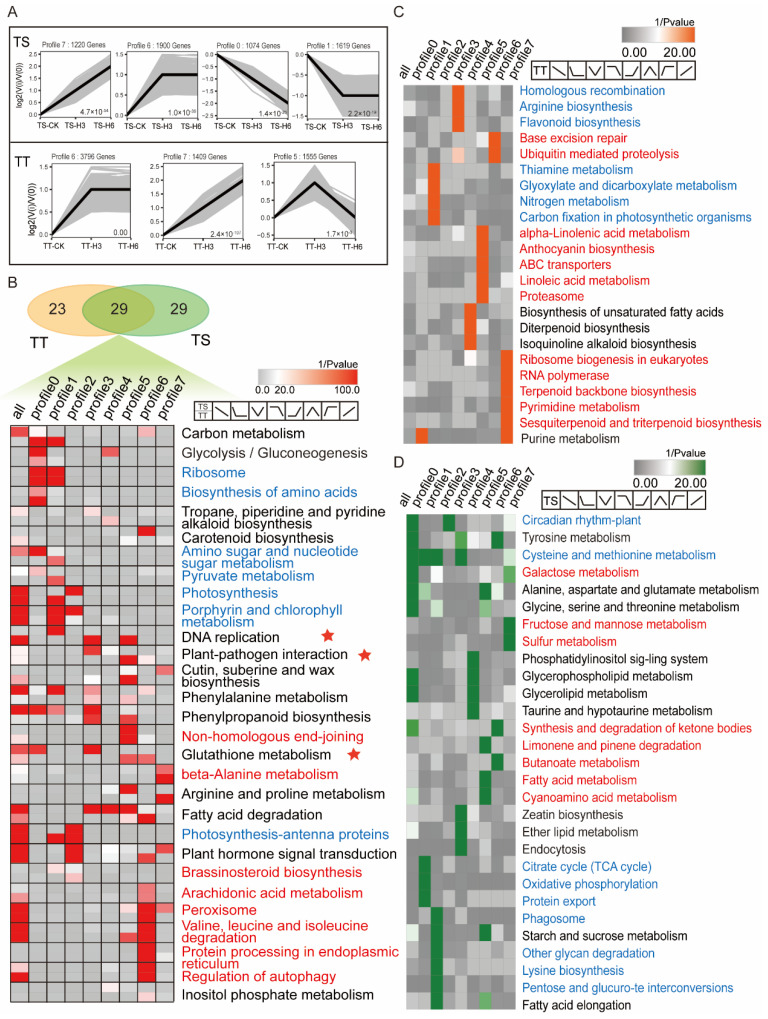
Clustering analysis and KEGG enrichment of DEGs based on the gene expression profiles. (**A**) The significant expression profiles based on time points in TS and thermo-tolerant TT cucumber lines after high-temperature stress. In each profile, each light gray line represents each gene, while the thick black line represents the global expression tendency, and the number on the right is the *p*-value. (**B**–**D**) Venn diagram for enrichment pathways (*p* < 0.05) of the two cultivars, presented as a heat map. In the heat map, the significant enrichment pathways of the two varieties are presented (**B**), and specific significant enrichment pathways are also shown for thermo-tolerant TT (**C**) and thermo-sensitive TS cucumber (**D**) varieties. Red and blue fonts indicate the pathways that were significantly enriched with upregulated or down-regulated genes in both varieties, while ☆ indicates that the significant enrichment pathways of thermo-tolerant and thermo-sensitive cucumber presented different expression patterns.

**Figure 5 ijms-23-01817-f005:**
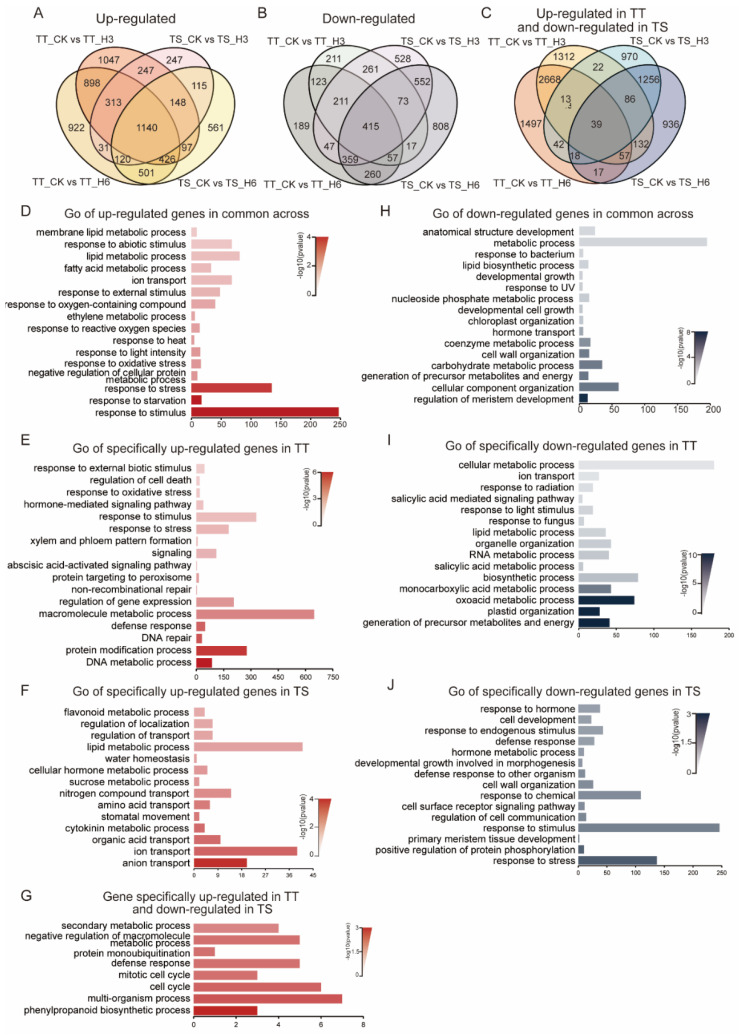
The differences of thermo-tolerant and thermo-sensitive cucumber line DEGs in response to heat stress analyzed by Venn clustering and GO. (**A**–**C**) Venn diagrams for common and specific heat responsiveness of different times in thermo-tolerant and thermo-sensitive cucumber lines. (**D**–**J**) GO analysis of biological processes affected by high-temperature stress. Heat upregulated the genes of different biological processes across the two lines (**D**), specifically upregulated in thermo-tolerant (**E**) and thermo-sensitive (**F**) cucumber lines. Biological processes were upregulated in thermo-tolerant but downregulated in thermosensitive cucumber lines following heat treatment (**G**). The genes of different biological processes were downregulated by heat across the two lines (**H**) and specifically downregulated in thermo-tolerant (**I**) and thermosensitive (**J**) cucumber lines. *p*-values are shown in different color shades. The column chart represents the number of genes in a biological process.

**Figure 6 ijms-23-01817-f006:**
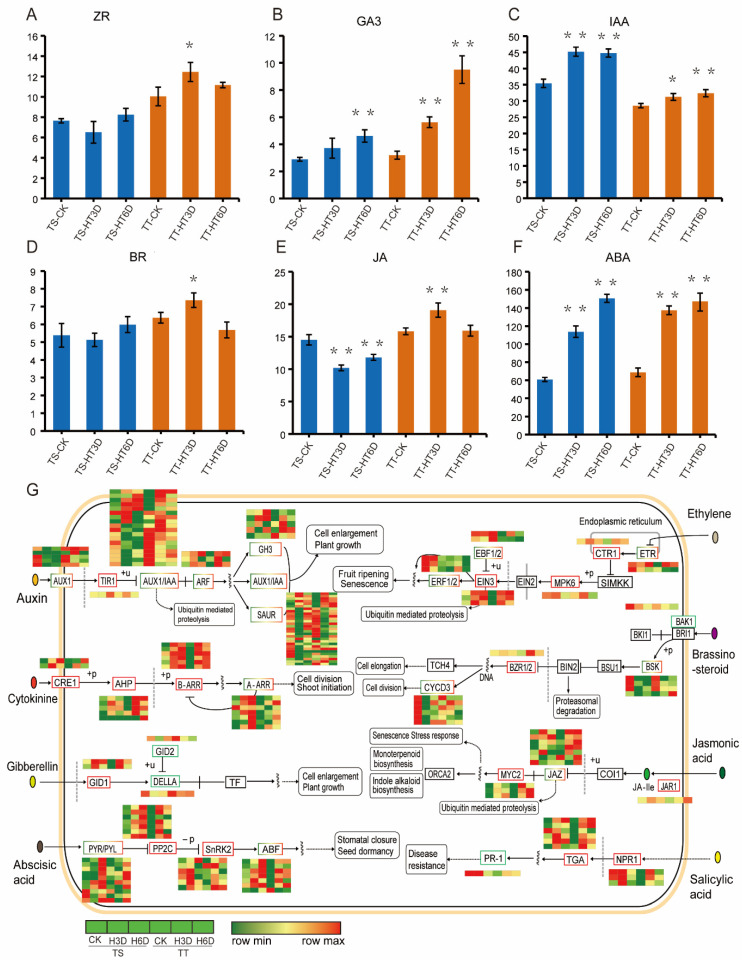
Changes in the endogenous hormone content and gene expression of plant hormone signal transduction pathways in thermo-tolerant and thermo-sensitive cucumber lines under high-temperature stress. (**A**–**F**) Changes in the endogenous hormone levels in thermo-tolerant and thermo-sensitive cucumber lines under high-temperature stress. ZR, zeatin, mean cytokinin; GA_3_, Gibberellin A3; IAA, Indole-3-acetic acid, mean auxin; BR, Brassinosteroids; JA, Jasmonic acid; and ABA, Abscisic acid. Error bars represent the mean ± SD. * and ** indicate significant differences from control plants (CK) at *p* < 0.05 and *p* < 0.01, respectively. (**G**) Plant hormone signal transduction pathways in thermo-tolerant and thermo-sensitive cucumber lines under high-temperature stress. Each row represents a significantly different expression gene. The maximum to minimum values of gene expression in the same row are given a corresponding color.

**Figure 7 ijms-23-01817-f007:**
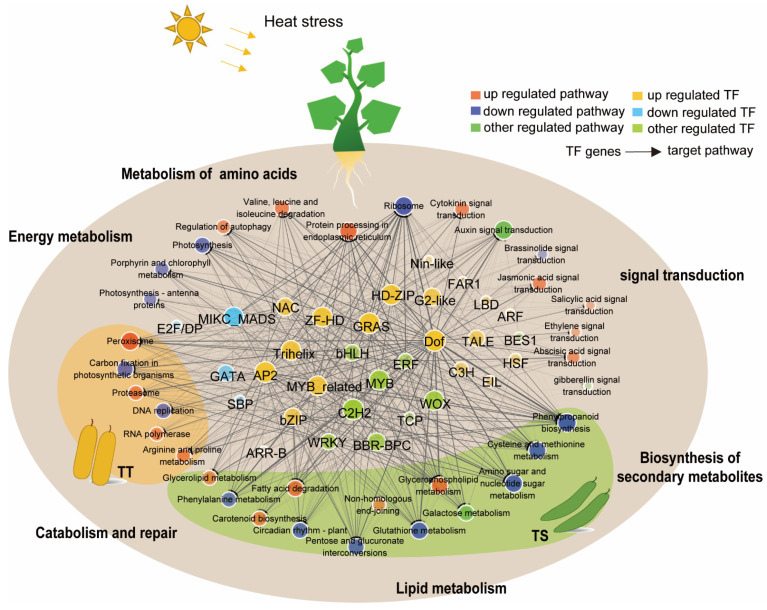
Possible interactions between TF and HT significant enrichment pathways in thermo-tolerant and thermo-sensitive cucumber lines under heat stress. Protein interactions and correlation degrees between TFs families and the HT significant enrichment pathway (q < 0.05) in thermo-tolerant and thermo-sensitive cucumbers under high-temperature stress. The predictive data were screened by statistical test (q < 0.05, FDR < 0.05) and correlation analysis (ρ > 0.9). The size of the nodes represents the degree of influence, the color of the nodes shows the overall expression trend of the enriched gene, while the width and color transparency of the edges represents the predicted scores. The significant enrichment pathway of TT is in the yellow region, the TS is in the green region, and the other regions are the common significant pathways.

## Data Availability

The raw data supporting the conclusions of this article will be made available by the authors without undue reservation.

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
