# Peer review of "Heat Stress Resistance Mechanisms of Two Cucumber Varieties from Different Regions"

_ijms, 2022, doi:10.3390/ijms23031817_

Round 1

Reviewer 1 Report

thank you for a paper that is quite readable

I have used sticky notes where there are comments.  There are places in methods where your description is lacking--  goes back to the plant physiologist that I am.  And the traditional view that methods should allow anyone to repeat the study.  

Some of your  figures do not explain the vast data sets you are conveying. In one even when enlarged you cannot read words --too blurred.  in others labels for axes are missing although some information is in the legend. 

What do you plan to do with the huge amount of data accumulated?  Does it have breeding value, GMO significance etc  or is it just knowledge and a way to show sophisticated data analyses?   ie busy work 

Do you plan to do the reverse  ie see how the S variety works in the N growth conditions  ? 

Author Response

Many thanks for your suggestions and comments. These had pointed out a lot of shortcomings for me. I have actively revised each of your comments and suggestions in the manuscript. Some of the images we didn't realize that were confusing to readers and didn't convey what we wanted to say, so we tried to fix them. Thanks much again for your advices.

The last question is great, and it's something we always think about when we do research. A huge amount of data is the basis of our subsequent studies. They have breeding significance, but in terms of verifying gene functions and increasing breeding gene resources, this manuscript is lacking, which is also the focus of our subsequent research. On the other hand, plant heat resistance is a very complex process, which not only affects various life activities in plants, but also makes plants more susceptible to the stress of drought, flood, saline-alkali and diseases, and the performance of different crops is very different. Therefore, we believe that a macroscopic understanding of the heat resistance of cucumbers will be helpful to our research and perhaps enlighten other researchers. Of course, there are still many shortcomings in our research, and I hope that we can continue to make progress.

Reviewer 2 Report

The paper investigated the heat tolerance of two cucumber varieties in order to elucidate some of the mechanisms involved in thermotolerance.

Generally, the paper is well written and deserves to be published. There are some minor points which could improved the paper:

In the abstract, the codifications with '14' and '2' are strange and could be removed.

The introduction is too long. The description of different cucumber varieties could be shorter.

The description of high-temperature stress is missing.

The transcription regulation is poorly described.

Author Response

Thank you very much for your recognition and advices. I have changed the troublesome varieties numbers ‘14’ and ‘02’ to thermotolerant (‘TT’) and thermosensitive (‘TS’). I added the description of cucumber heat damage in the introduction and simplified the original content in the introduction. Transcription in hest stress is a very complex and imperfect content. Especially in my manuscripts, the transcription regulation part is really not good enough, I also made some modifications. Thank you very much for your advice, which is very helpful for me to improve the manuscript.
